# An index of access to essential infrastructure to identify where physical distancing is impossible

Isabel Günther[1,2], Kenneth Harttgen [1,2✉], Johannes Seiler [2,3] & Jürg Utzinger[4,5]

To identify areas at highest risk of infectious disease transmission in Africa, we develop a physical distancing index (PDI) based on the share of households without access to private toilets, water, space, transportation, and communication technology and weight it with population density. Our results highlight that in addition to improving health systems, countries across Africa, especially in the western part of Africa, need to address the lack of essential domestic infrastructure. Missing infrastructure prevents societies from limiting the spread of communicable diseases by undermining the effectiveness of governmental regulations on physical distancing. We also provide high-resolution risk maps that show which regions are most limited in protecting themselves. We find considerable spatial heterogeneity of the PDI within countries and show that it is highly correlated with detected COVID-19 cases. Governments could pay specific attention to these areas to target limited resources more precisely to prevent disease transmission.

[1] Development Economics Group, ETH Zürich, Zürich, Switzerland. [2] NADEL - Center for Development and Cooperation, ETH Zürich, Zürich, Switzerland. [3] Department of Statistics, University of Innsbruck, Innsbruck, Austria. [4] Swiss Tropical and Public Health Institute, Allschwil, Switzerland. [5] University of Basel, Basel, Switzerland. ✉email: kenneth.harttgen@nadel.ethz.ch

Limited infrastructure remains a challenge for many low- and middle-income countries (LMICs), especially those in Africa. At the macro-level, inadequate infrastructure such as electricity and roads, is a major obstacle to achieving sustained economic growth in the region[1]. At the micro-level, millions of households lack access to basic private infrastructure such as clean water, sufficient living space, and improved sanitation[2]. Lack of such infrastructure also affects the ability of countries and people to prevent outbreaks and contain the spread of infectious diseases such as the ongoing coronavirus disease 2019 (COVID-19) pandemic[3].

As of the end of December 2021, there were more than 288 million confirmed cases of severe acute respiratory syndrome coronavirus-2 (SARS-CoV-2) globally, the causative agent of COVID-19, and more than 5.4 million people had died. The first confirmed cases on the African continent were reported on February 14, 2020 in Egypt and on February 15, 2020 in Algeria[4]. By the end of December 2021, more than seven million people had been diagnosed in Africa, with the bulk of cases concentrated in South Africa (3,458,286), Ethiopia (415,443), Kenya (292,237), Nigeria (241,513), and Ghana (141,295). On the African continent fewer laboratory-confirmed and fatal cases had been reported than in other regions of the world[5]. This is partly a reflection of the lack of testing facilities, which is still limited in many African countries[6–8]. For example, excess mortality has been shown to be higher than confirmed COVID-19 deaths[9,10]. Potential unobserved high infection rates can partly be explained by the lack of vaccination in Africa, with only 9.2% of the population in Africa being fully vaccinated whereas globally already 48.5% of the population has been fully vaccinated[11,12]. Hence, containing the spread of the virus as well as decreasing the risk of future variants will, in the short term, strongly depend on the progress of national and global vaccination campaigns. In the long-term, it will depend on investments in infrastructure that allow households to reduce their risk of getting infected in the first place—otherwise poorer countries may once again have to resort to lengthy school closures, which are easiest in the short term, but have devastating effects in the long term[13,14].

As recommended by the World Health Organization (WHO) at the beginning of the pandemic in 2020, most countries responded to the COVID-19 crisis with a series of public health policies until treatments against COVID-19 and vaccinations became available in 2021. In particular, regulations on mask wearing and recommendations for physical distancing (at least 1 or 1.5 m distance between two persons) in order to slow down the infection rate among the population were implemented globally[15]. To achieve physical distancing of their populations, governments have also enforced the suspension of schooling, movement restrictions, prohibition of gatherings, work-from-home regulations, and night-time curfews. Whereas in the Americas governmental regulations for physical distancing were put into place comparatively late and reluctantly, this was not the case for African countries. Indeed, most African governments decided to impose lockdowns of public life much earlier (in terms of confirmed COVID-19 cases) than most high-income countries (HICs). For example, by March 31, 2020, all African countries except Burundi had closed their schools. At that point in time, some African countries had zero confirmed cases. Important underlying reasons for early and often stringent governmental regulations were weak health systems and lack of medical infrastructure capacity to handle severe COVID-19 cases. In addition, even HICs such as Italy or the United States of America faced severe constraints in providing sufficient treatment possibilities to intensive care patients diagnosed with COVID-19.

In this work, we develop a physical distancing index (PDI) based on the share of households without access to private toilets, water, space, transportation, and communication technology and weight it with population density. Our findings reveal that many subnational regions in Africa have severe infrastructure constraints that undermine physical distancing policies at the household-level, which could lead to growing infection rates despite costly national measures, such as school, business, and border closures. Countries with the highest risk of rapid spread of disease transmission due to lack of essential infrastructure are countries in the western part of Africa. The findings also show large within-country heterogeneity for the PDI, pointing to high-risk areas within countries. In addition, the results highlight the fact that different countries face different infrastructure challenges. Finally, we find that the PDI is highly correlated with detected COVID-19 cases. Our approach allows more precise targeting of policy interventions such as vaccination campaigns and infrastructure investments, a strategy that many high-income countries also followed through their measures to identify hotspots of new COVID-19 infections through contact tracing. Given limited resources, effective targeting seems to be even more important for LMICs.

## Results

**Motivation and objective.** Measures of physical distancing in African countries came with immense economic costs. All countries in Africa are categorized as low-income (US$ 1025 or less per year and capita) or middle-income (US$ 1026 to US$ 12,375 per year and capita) as defined by the World Bank[16]. Therefore, these countries have very limited financial resources to mitigate any negative economic effects both at the macro- and the micro-level. The International Monetary Fund estimates that sub-Saharan Africa's gross domestic product (GDP) shrunk by 1.9% in 2020[17], which will result in a sharp increase in poverty[18,19] for the first time in 30 years. Estimates from the United Nations Development Programme indicate a sharp reduction in the Human Development Index in 2020 for the first time since its introduction in 1990[20]. Large shares of the populations are employed in the informal sector, with estimates varying between 35% for South Africa and 92% for Mali, with no social security net[2]. If people cannot go to work, the result is an instant income loss for most of these people, leading to an immediate rise in food insecurity[21,22]. As a result, many African countries quickly started to lift measures of physical distancing in summer 2020. However, schools remained closed in most African countries throughout 2020 and 2021[23,24].

Our study aims to contribute to a deeper understanding of the geographic distribution of critical infrastructure patterns to respond to the current and future epidemics and pandemics, placing particular emphasis on Africa. Our study also contributes to measuring a country's preparedness to prevent, detect, and cope with infectious disease outbreaks such as COVID-19[25–30]. We argue that the effectiveness of governmental regulations in many African countries to increase physical distancing and to reduce transmission rates of infectious diseases does not only lead to poverty but is also limited given the lack of essential private infrastructure, which makes it impossible for populations to follow WHO regulations to keep sufficient distance. Although vaccinations and treatments against COVID-19 became available in 2021, international and national barriers toward high vaccination coverage in many African countries will remain and these have also been discussed as a driver of future mutations of SARS-CoV-2[31]. Hence, to both contain the spread of SARS-CoV-2 and future viruses governmental measures to encourage physical distancing remain important policy responses.

**Index of access to essential infrastructure**. Using principal component analysis, we propose a physical distancing index (PDI) composed of five indicators: households with (1) a lack of private toilet facilities; (2) lack of a private drinking water source; (3) lack of ICT infrastructure; (4) lack of private transportation means; and (5) lack of space. The indicator is weighted with population density to account for the fact that the capacity to keep physical distance is both influenced by the lack of private infrastructure and population density. We compute the PDI for 34 African countries as well as for 519 first-level subnational regions. Moreover, based on Bayesian distributional regression, the PDI is computed at the pixel level (grid size of $5 \times 5$ km) for specific countries.

**Comparisons to existing indices**. The proposed index complements existing indices that have attempted to measure a country's capacity to respond to an infectious disease outbreak. Most existing indices focus on measuring the overall capacity of the country's health and governance system to detect and respond rather than on households' capabilities to prevent the spread of an infectious disease through physical distancing. For example, one attempt to measure the preparedness of a country's health system to deal with an infectious disease outbreak is the monitoring of the International Health Regulations (IHR) by the WHO[25]. The aggregated index to monitor progress in a country's health system was introduced in 2010 and is based on 13 different capacity dimensions: (1) legislation and financing; (2) IHR coordination and national IHR focal point functions; (3) zoonotic event and the human-animal interface; (4) food safety; (5) laboratory; (6) surveillance; (7) human resources; (8) national health emergency framework; (9) health service provisions; (10) risk communication; (11) points of entry; (12) chemical events; and (13) radiation emergencies. The most recent data from the year 2018 show a global improvement across all 13 IHR capacity dimensions. However, countries in Africa lag behind most other countries in the world[25]. A second index to analyze the vulnerability of countries with respect to infectious disease outbreaks is the Infectious Disease Vulnerability Index, developed by the RAND Corporation. The aggregated index is based on seven dimensions of factors influencing a country's vulnerability to infectious diseases: (1) demographic; (2) health care; (3) public health; (4) disease dynamics; (5) political-domestic; (6) political-international; and (7) economic[26]. The estimates of the index in 2016 show that of the 25 most vulnerable countries, 22 are in Africa (the other three are Afghanistan, Haiti, and Yemen). Particular disease hotspots are identified in West Africa, and the authors of the study point to a dangerous mix of political instability and limited capacity of health systems in countries such as Somalia, Central African Republic, and South Sudan[26].

The results of these two indices are limited to country-level aggregates and provide no within-country variation. Although estimates at the country level are useful for international and inter-temporal comparisons, they do not provide any information on within-country heterogeneity in preparedness to contain a disease. At the subnational level, where differences in policies and behavior within a country are less severe than across countries, a subnational PDI can be used for a more precise monitoring and targeting of outbreaks of infectious diseases. Moreover, the two indices provide no estimate on how the spread of infectious viruses, for example of the SARS-CoV-2 Delta and Omicron variants, can be contained through physical distancing practiced by the general public. Here, Brown et al.[29] provide a first attempt, but our approach differs in four fundamental dimensions. First, Brown et al. study different indicators: (1) household has access to internet, phone, TV, or radio; (2) no more than two people per

sleeping room; (3) household has access to a private toilet; (4) household has a dwelling that can be closed; (5) household has access to piped water; and (6) household has a place for handwashing. We focus on indicators that are more directly linked to social interaction: for example, whether a household has a TV or a place for handwashing says little about social interaction. Second, we exploit the availability of geo-referenced information in the Demographic and Health Surveys (DHS) to provide new insights about the capability to physical distance at the subnational level, whereas Brown et al. only aggregate at the national level. Geo-referenced data can help to identify potential diseases hotspots within a country for better policy targeting. Third, Brown et al. use a simple country average of their indicators to calculate their index. Although this is a straightforward approach, it also implies an arbitrary weighting scheme where one has to assume that, for example, access to a TV has the same informative power as sharing a room in explaining the capacities of households to protect themselves from getting infected. We employ the PCA method to avoid the equal weighting assumption, which is a commonly used approach in the empirical literature. The PCA is a more data-driven approach and combines the variation of all included variables in the index. Fourth, we take into account population density, which we argue is critical in studying the capabilities of households to physically distance, as higher population density is associated with higher infection risk when private infrastructure is lacking (see also Fig. 3). As a result of all these differences, the correlation between the home environment for protection index (HEP) and our PDI is very low ($\rho = 0.2$, see also Fig. 5), also resulting in a different ranking of countries with respect to their capability to distance physically.

While our results show some similarities to the results of existing indices that measure the functioning of a country's health system and the vulnerability of countries with respect to infectious disease outbreaks, our results also show some interesting differences. For example, Ghana and Senegal are, relative to other African countries, ranked high in the existing indices; however, due to their high population density and limited private infrastructure, the risk of disease transmission is still high. Furthermore, some countries even show a double burden of a high PDI (very limited capability to keep physical distance) and a low capacity of the health system to deal with an outbreak of an infectious disease, such as Benin, The Gambia, Sierra Leone, and Togo.

**Limited essential private infrastructure**. Figure 1 depicts the results of the geospatial estimates of the population weighted PDI at the country and regional level for all 34 countries in Africa for which we have data. A higher index value and darker color represent a lower capability to physical distance, and hence, a higher risk of disease transmission. The corresponding country average values for each indicator of the PDI as well as the normalized index value are presented in Supplementary Table 2.1. Moreover, Supplementary Fig. 1.2 shows the population density at the country and regional level and disaggregated at the pixel level. As expected, countries with a high population density show an increase in the index (or a decrease in the capability for physical distancing) when adjusting by population density.

We find considerable heterogeneity in the PDI across Africa. High-risk areas of disease transmission are particularly concentrated in the western part of Africa, such as Ghana, The Gambia, Togo, Sierra Leone, Benin, Liberia, Senegal, and Côte d'Ivoire. A relatively high population density (for example Ghana, The Gambia, and Togo had population densities between 121 and 200 people per km$^2$ in 2015), coupled with limited infrastructure for

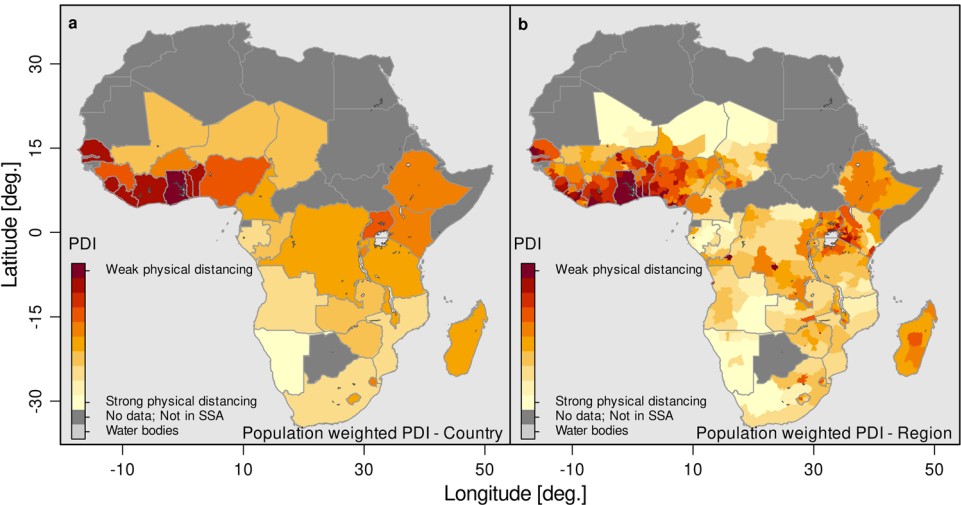

**Fig. 1 PDI at country and regional level.** Country (**a**) and regional (**b**) level PDI. The panel depicts the capabilities of households to follow social distancing measures based on a simple multidimensional measure calculated based on (1) number of households sharing toilet facilities, (2) usage of public water source, (3) persons per room, (4) no access to ICT, (5) bicycle or other vehicle is not present, based on a PCA. The estimates are normalized between zero and one. Source: DHS and Center for International Earth Science Information Network—CIESIN—Columbia University[52]; calculations by the authors.

physical distancing, could make these countries highly susceptible to infectious diseases that are transmitted through droplets. Countries with lower population densities and relatively better essential private infrastructure, such as Namibia, Gabon, Mozambique, and South Africa, show (relative to other countries in Africa) a lower PDI. Figure 1 also shows that countries such as Niger and Chad, despite facing a severe lack of infrastructure, might still face slower transmission rates compared to countries with an equally severe infrastructural challenge, such as Liberia or Ghana, due to lower population densities. The interpretation of the geospatial estimates need to be made in relation to other African countries. For example, although South Africa shows a much brighter color in Fig. 1, this does not mean that South Africa has all the infrastructure in place for people to keep distance, in particular in socio-economically deprived and marginalized settings. Moreover, even if such infrastructure is in place, it does not mean that people necessarily follow physical distancing recommendations[32].

More interestingly, Fig. 1 (right panel) also reveals considerable spatial heterogeneity of high-risk areas within countries. Whereas some countries show a relatively consistent risk pattern, such as Sierra Leone, Liberia, and Ethiopia, other countries reveal hotspots within countries that are hidden in the estimates of the national average. For example, western Kenya is a very high-risk region (Kisumu, Mombasa, and Nairobi), as is southern/central Côte d'Ivoire (Abidjan, Bas-Sassandra, and Yamoussoukro), north-western Tanzania (Geita, Shinyanga, Simiyu, and Tabora), or north-east South Africa (KwaZulu-Natal and Gauteng). Hence, although country-level estimates are useful for international or inter-temporal comparison, they mask important differences in the risk of disease transmission due to lack of infrastructure at lower administrative levels. This is pivotal to prioritizing national interventions, such as increased testing efforts or vaccination campaigns in the most vulnerable regions of countries.

Figure 2 shows the results of the Bayesian regression at the pixel level for Ghana, Ethiopia, Kenya, and South Africa, four countries with some of the highest numbers of SARS-CoV-2 cases in Africa registered as of August, 2021. These countries are ranked amongst the least (South Africa), the middle (Ethiopia and Kenya) and the most (Ghana) challenging in the infrastructure-based PDI. For all countries, subnational

heterogeneity is high and high-risk areas exist in all countries where people cannot protect themselves by keeping distance and are, hence, highly susceptible to the spread of infectious diseases by droplets. Moreover, in these areas lockdowns of public life will be difficult to enforce as people will have to leave the house not only to buy food and access health services, but also to access other public infrastructure.

To assess whether the PDI indeed hints to potential hotspots of disease transmission, we checked all countries in our sample to see if data on reported COVID-19 cases is available at the subnational level, and identified nine countries with subnational regional information. The countries are: Democratic Republic of the Congo, Ethiopia, Mozambique, Namibia, Nigeria, Niger, Senegal, South Africa, and Togo. Figure 3 illustrates the close association between the PDI and number of COVID-19 cases for South Africa. The comparisons of PDI and COVID-19 confirmed cases for the other eight countries are shown in Supplementary Figs. 1.3 and 1.4. The correlation coefficient between the PDI and COVID-19 cases for all nine countries ranges from 0.4 to 0.9, pointing to an overall close association between our PDI index and the observed regional caseload. This simple ex-post comparison provides evidence about the predictive power of the PDI to identify potential disease hotspots within countries.

A closer analysis of the different indicators entering the PDI (see Fig. 4) helps to explain the occurrence of hotspots and provides guidance to countries where infrastructure investments are most needed. Different countries face different challenges. For example, Ghana and Liberia have severe private sanitation constraints, Rwanda and Burundi face severe private water infrastructure constraints, The Gambia and Senegal show more crowded housing, and the populations of Madagascar and the Democratic Republic of the Congo, do not have access to private communication or transportation. Across the African continent, 45% of households share toilets. On average, households share toilets with two other households, but the average number ranges from 1.32 households in Mozambique to 6.17 households in Ghana. To expect these households not to meet other people on a regular basis is simply unrealistic. The average number of people per room is 3.2, showing the difficulties of households and families to effectively isolate if a household member becomes sick. In Senegal and The Gambia this number goes up to five people

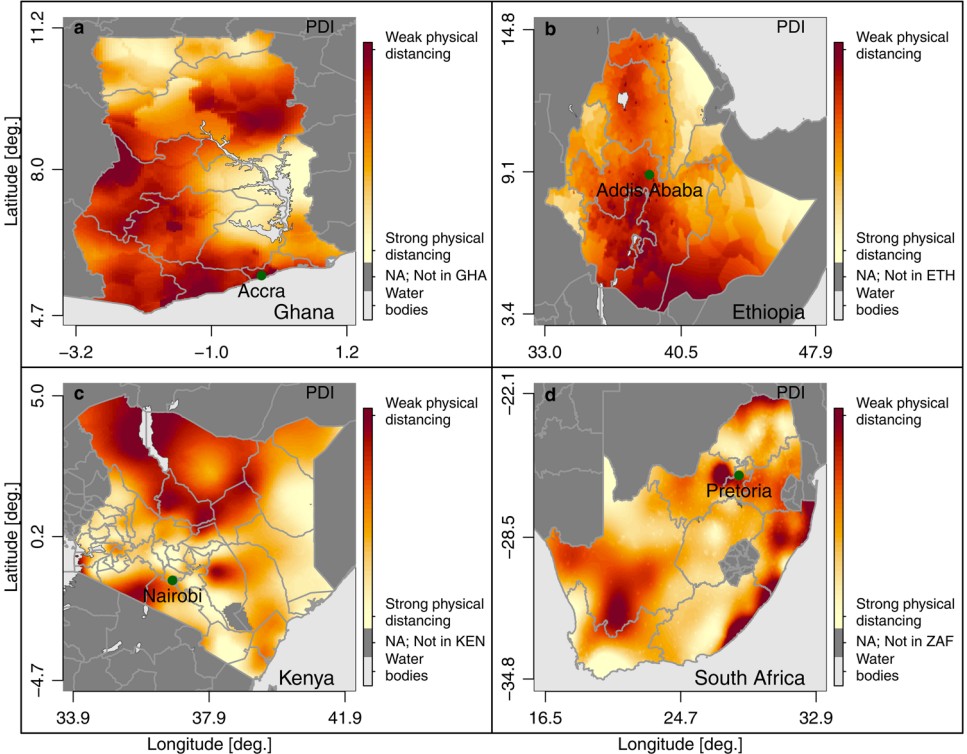

**Fig. 2 PDI at 5 × 5 km level for Ghana, Ethiopia, Kenya, and South Africa.** Estimates of the PDI at pixel level (5 × 5 km) for Ghana (**a**); Ethiopia (**b**); Kenya (**c**) and South Africa (**d**). Source: DHS and Center for International Earth Science Information Network—CIESIN—Columbia University[52]; calculations by the authors.

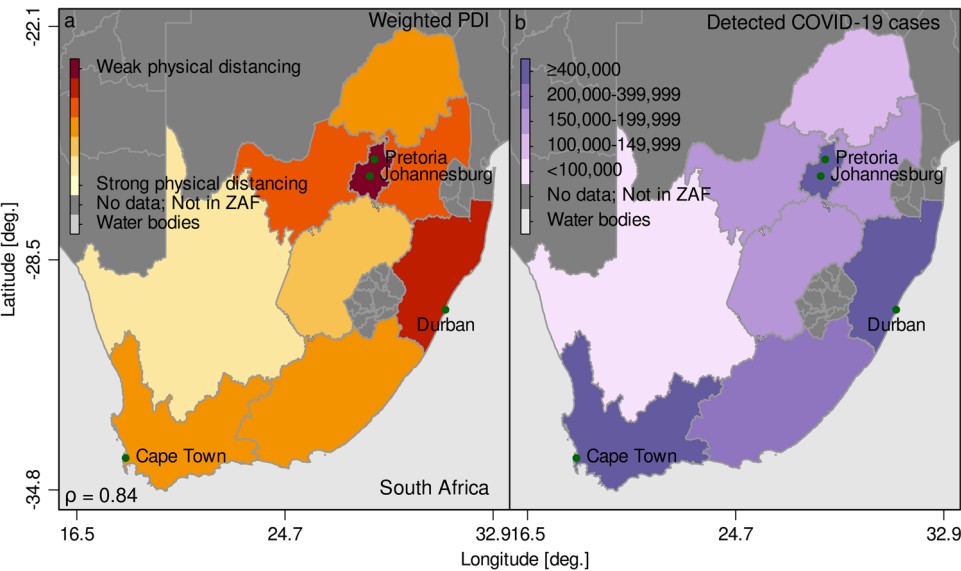

**Fig. 3 Comparison of the PDI and detected COVID-19 cases in South Africa.** Population weighted PDI (**a**), and observed cumulative caseload (**b**) at the regional level for South Africa. Note that we used the latest available information on the aggregated regional caseload. Source: Information on the regional caseload for South Africa is publicly available data which we are happy to share upon request and DHS; calculations by the authors.

sharing the same room for sleeping. Shared sanitation and sharing a room with many other household members are the two indicators with the highest weight in the PCA (see Supplementary Fig. 1.7), meaning that regions and countries that show severe infrastructure constraints in access to a private sanitation facility and private room show the highest PDI values. For 40% of the households in our sample, the only access to water is from a public water source; these households need to leave their house to gather water, which increases the risk of infection. The share of households that do not own a mobile phone ranges from 56% in Madagascar to 3% in Senegal. Similarly, the share of households that own a bicycle, motorbike or car ranges from 5% in Ethiopia to 94% in Burkina Faso, again emphasizing the high heterogeneity between countries.

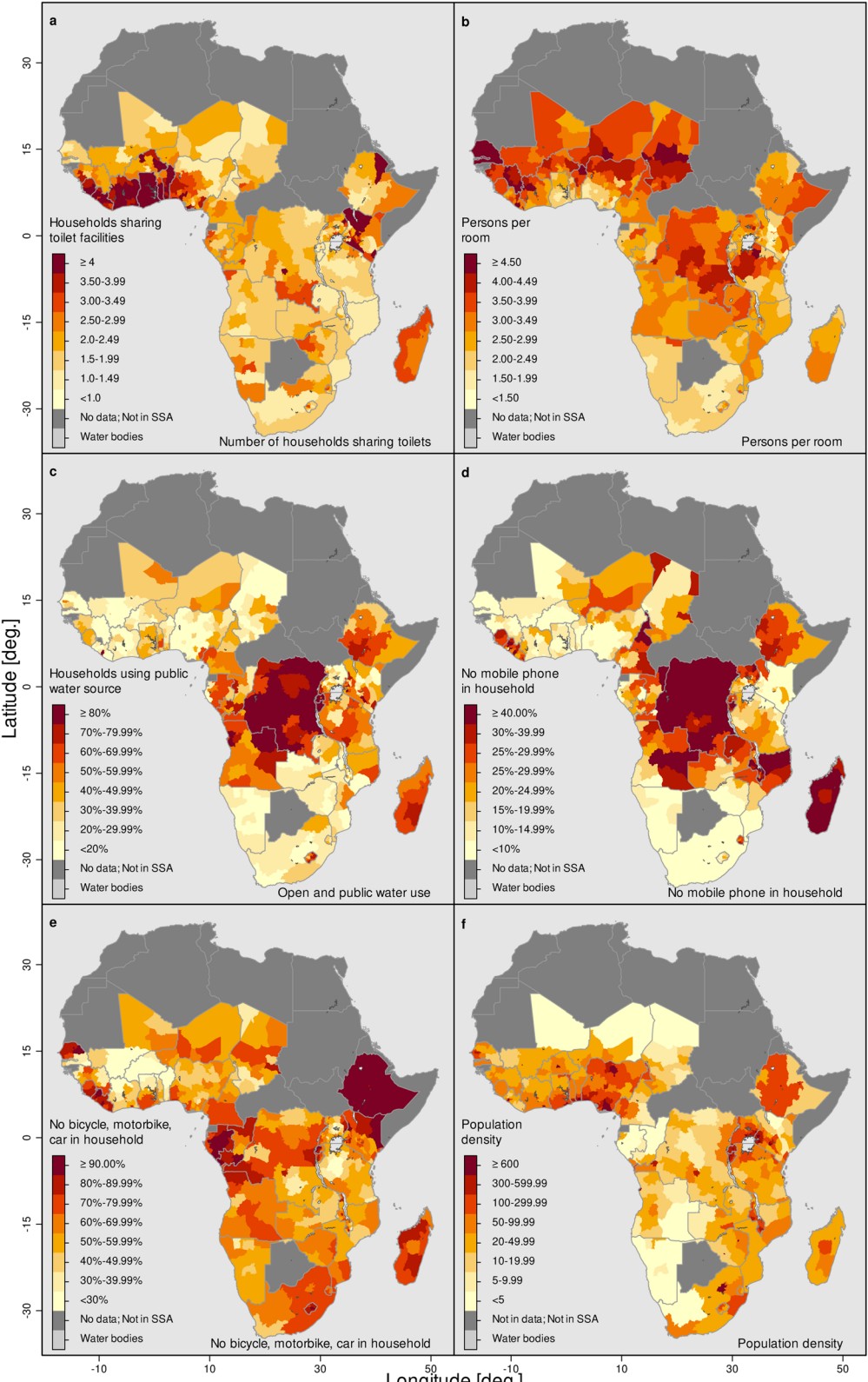

**Fig. 4 Indicators of the PDI and population density at the regional level.** The figure shows the shares at the regional level from top left to bottom right: (**a**) number of households sharing sanitation facilities, (**b**) number of people per room, (**c**) share of households using open and public water sources, (**d**) share of households without a mobile phone, **e** share of households with no bike, car or motorbike, and (**f**) population density (people per km$^2$) estimates for 2020. Source: DHS, and Center for International Earth Science Information Network—CIESIN—Columbia University[52]; calculations by the authors.

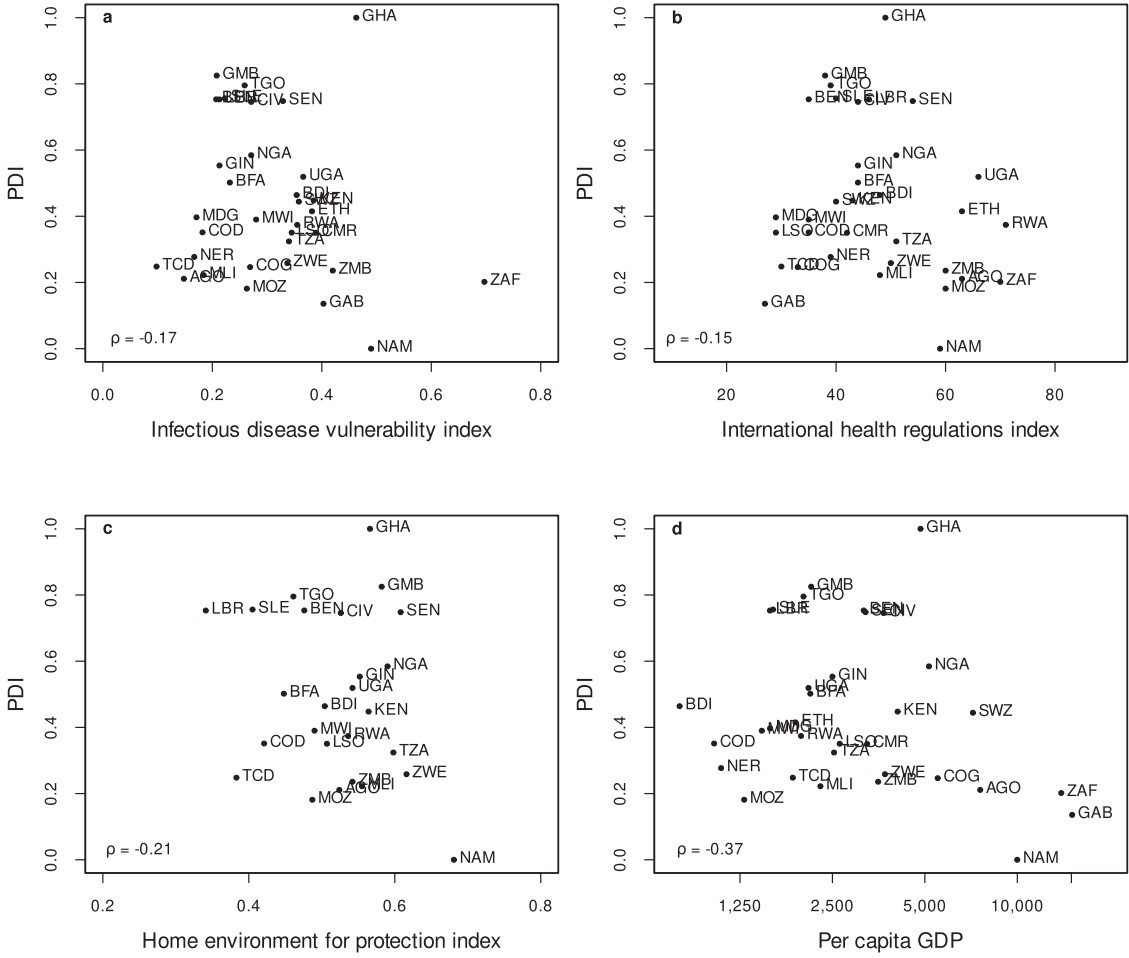

**Fig. 5 Correlation of the PDI with related indices and GDP per capita.** Scatter plot of the PDI (using the latest available Demographic Health Survey of a country) against the infectious disease vulnerability index (**a**)[26]; the WHO international health regulations index (**b**)[25]; the home environment for protection index (**c**)[29]; and the country's GDP (**d**)[2]. Note that a high PDI implies a lack of essential private infrastructure, and hence, high risk of disease transmission; accordingly, countries in the top of boxes have a lack of private infrastructure, and hence, lack the capacity to limit disease transmission. Moreover, note that for the infectious disease vulnerability index, the WHO international health regulations index, and the home environment for protection index, a higher value indicates a better preparedness, or protection. In addition, ISO-3 country codes are used to abbreviate the countries in this figure. See Supplementary Table 2.1 for details on the ISO-3 country code. Source: DHS, Moore et al.[26], Gilbert et al.[28], and World Development Indicators[2]; calculations by the authors.

Comparing the results of the PDI to other indices that measure a country's vulnerability to a pandemic outbreak or the general ability of a health system to deal with an outbreak shows a weak correlation between the indices in general (see Fig. 5). Countries such as Benin, The Gambia, Sierra Leone, and Togo have both weak health systems to deal with a sudden outbreak of an infectious disease, as well as a lack of essential private infrastructure—such as access to private water, toilets, transportation, ICT, and space—that undermine measures aimed at slowing the spread of a pandemic. Countries like Ghana face severe infrastructural constraints to slow down the spread of a pandemic such as COVID-19, but have the capacity of the national health system to respond to it. Countries like Rwanda, South Africa, and Namibia have both a functioning health system to respond, and access to essential private infrastructure to facilitate COVID-19 prevention measures. Case numbers in South Africa have still been the highest in Africa, which shows that infrastructure is not sufficient and countries heavily depend on their people to adhere to public health measures, such as physical distancing[32]. The high general caseload in South Africa can also be seen as a reflection of the high number of tests relative to other countries in Africa[11,12], the higher importation risk of

COVID-19[28], and the older population[33]. Hence, the PDI does not, on its own, provide any predictions about outbreaks on the national level but can help to identify regions within a country where infectious diseases might spread faster once they enter these regions (see Fig. 3 and Supplementary Figs. 1.3 and 1.4). Last, we observe only a weak negative association between the PDI and per capita GDP (Fig. 5), indicating that economically more advanced countries have a lower risk of disease transmission. However, the relationship seems to be non-linear and, particularly for poorer countries, heterogeneity seems to be high. This means that despite being poor, some countries have managed to provide basic essential infrastructure, which helps to protect their populations and improves their livelihoods. This argument is further emphasized by Supplementary Fig. 1.5, which plots the PDI against the regional poverty headcount rate at the first administrative level for countries with available data, indicating that simply targeting poor regions with intensified COVID-19 prevention measures is not sufficient.

## Discussion

The ability of a government to respond to an infectious disease outbreak[25–28] and the collective behavior of the population[32] are

key to containing the negative consequences of an emerging disease. We argue that in addition, the proportion of the population with access to essential private infrastructure has an impact on the ability of societies to protect themselves against infectious diseases.

The spatial analysis shows that, in general, many households in Africa lack essential private infrastructure to foster physical distancing. If not addressed in the long-term, infectious diseases, like COVID-19, will continue to spread despite drastic and costly national measures—such as closure of schools and businesses—that are intended to reduce public contacts. Our results also point toward the requirement of alternative policy approaches in the short term to prevent the spread of infectious diseases in these countries. First, the international community needs to increase the delivery of basic safety measures to countries at risk, such as masks and hand sanitizers (most importantly soap and washing facilities at community toilets, which would have ancillary benefits, such as reducing diarrheal diseases). Second, vaccinating populations that do not have the capability to keep distance in their daily lives should become the priority of the international community. However, to date, only about 125 million people in Africa have been vaccinated, whereas this number is already 459 million in Europe and 3817 million globally as of the end of December 2021[11,12,34].

Our research also allows the identification of hotspots of the PDI in countries where infectious diseases are more likely to spread. These maps are particularly relevant to tailor surveillance-response systems in a spatially explicit manner to maximize the impact of scarce fiscal resources in LMICs. Our results might, hence, inform national programming to help mitigate the spread and effects of potential regional and national outbreaks. In addition, our approach can provide a basis for a more general framework to be applied for future outbreaks of other infectious diseases, though context, disease agent, transmissibility, and human behavior will also play important roles.

Limitations of our study include lack of information beyond private water, sanitation, space, communication and transport as well as population density, and no information about the probability of a virus entering the country or effects of policy measures to contain COVID-19[35]. Most importantly, information about the behavior of the population within and across countries is lacking. This causes a discrepancy between the PDI and the observed caseload at the national level and the PDI can therefore not be used on its own as a prediction tool for outbreaks at the national level. However, we have shown that the PDI can be used to identify regions within a country where infectious diseases might spread faster once they enter these regions. Furthermore, small-scale heterogeneity, for example within cities (e.g., on a 1 × 1 km grid) cannot be assessed using DHS data, given that GPS coordinates are randomly displaced to guarantee anonymity. Last, recent data for some of the poorest countries in Africa (e.g., South Sudan, Somalia, Central African Republic, and Somalia) is not available.

To conclude, containing the spread of the virus as well as decreasing the risk of future variants will, in the short term, strongly depend on the progress of national and global vaccination campaigns. In the long term, it will depend on investments in infrastructure that will allow households to reduce risks of getting infected in the first place. Our findings indicate a low ability of households in most African countries to protect themselves from contracting infectious diseases, such as SARS-CoV-2, because the essential infrastructure for daily life has to be shared. These vulnerable populations should get first priority in vaccination campaigns together with vulnerable demographic population groups (such as the elderly, the sick, or people working in health care). Moreover, within almost all African countries, even in

those with generally higher levels of private infrastructure, hotspots with almost no infrastructure remain. Although this lack of infrastructure becomes particularly apparent during a pandemic—such as COVID-19—it has also limited the well-being of the population in non-crisis times. It is essential to address this infrastructure crisis both to improve the ability of countries to implement public health measures in the event of future pandemics as well as to improve the quality of life for people in Africa.

## Methods

**Data collection**. Data for the analysis were taken from the Demographic and Health Surveys (DHS), which are administered by ICF International. The DHS are nationally representative cross-sectional surveys that have been conducted in over 80 LMICs at varying intervals starting in 1985, and are still ongoing. The DHS are designed to collect nationally representative health and welfare data of women of reproductive age, their children, and their households. A key advantage of the DHS is the availability of comparable data for multiple countries and the consistent quality of reporting and data over time[36]. We employed the largest available, nationally representative, and mutually comparable repeated cross-sectional samples from African countries for the most recent available survey year, which is between 2016 and 2018 for most countries, and no earlier than 2007. Supplementary Fig. 1.1 shows (for countries for which we have more than 1 year of data) that the PDI has—unfortunately—not changed considerably for almost all countries over the last 5–10 years, i.e., it is fairly stable over time. Hence, countries for which we have very recent data can be compared with countries for which the last available survey year is somewhat older.

We focus on five pieces of private infrastructure needed to enable households to comply with physical distancing measures: (1) a lack of private toilet facilities; (2) lack of a private drinking water source; (3) lack of ICT infrastructure (measured as a binary indicator capturing whether a mobile phone is available in the household); (4) lack of private transportation means; and (5) lack of space (measured as people per room used for sleeping). Access to these basic services remains limited in many LMICs, both in rural areas as well as in densely populated informal urban settlements. Shared sanitation facilities are of particular concern[37–39]—not only because one has to leave the house frequently to use them, but also because of worse hygienic sanitation conditions[40–42], which is closely related to the spread of diseases[43–45]. It is important to differentiate between indoor and outdoor shared spaces, especially for the COVID-19 pandemic, for which it has been shown that transmission indoors is much higher than transmission outdoors (with the same protective measures). See for example[46–48]. For our indicators, we can differentiate between indicators focusing on indoor and outdoor activities: water sharing is most likely outdoor, whereas everything else is indoor; even though two people would not be simultaneously in one shared sanitation space, sanitation could also be considered to be outdoor. The number of people in the household sleeping in the same room captures the within-household limitation for distancing and isolation in case a household member falls sick. Compared to other respiratory diseases like SARS, or the Middle East respiratory syndrome, the transmission incidence within households is higher for COVID-19[49]. In addition, we include whether the household has access to means of communication, which measures the connectedness of the household to the outside world. Owning, for example, a mobile phone is important to access news, information, and to learn about the risks of COVID-19. Last, Durizzo et al.[32] have shown that using public transportation represents the biggest obstacle to keeping social distance for people in low-income countries.

Focusing on these variables, we have data for 34 African countries from more than 700,000 households (see Supplementary Table 2.1 for the complete list of countries as well as for the country averages of the indicators entering the index taken from the latest available survey).

**Overview**. The proposed PDI for each country and region is based on a PCA, as suggested by refs. [50,51] for the construction of a wealth index based on the household's durables. First, we utilized PCA to construct an infrastructure index for each household. The main idea of this approach is to construct an aggregated one-dimensional index over the range of variables related to the households' ability to maintain physical distance, where the first principal component is the PDI. See Supplementary Fig. 1.7 for the variance of the first component of the PCA and the effect size of each variable used to obtain the overall PDI. A detailed description of the method is provided in the Supplementary Information. The included variables are selected based on the WHO recommendations for containing the spread of SARS-CoV-2. Supplementary Fig. 1.6 shows that the index does not change considerably if we exclude any of the five indicators of private infrastructure. Hence, no included indicator of private infrastructure is driving our results. The unweighted PDI is then aggregated over the country and regional (admin-1) levels, respectively, using the household sample weight provided in the household member recode of the DHS.

In a second step, we account for the fact that a lack of private essential infrastructure leads to more social interaction in more densely populated areas. For

that purpose, the household-level DHS data are merged with remotely-sensed information on population density from the Socioeconomic Data and Applications Center[52]. We use these data to achieve a weighting of the PDI by incorporating population density in the aggregation process, where the household-level data are aggregated to the country and regional (admin-1) level using $log(1 + population density)$ as an additional weight in the aggregation process. Population density is measured as people per $km^2$.

In an additional step, we apply Bayesian simulation techniques to provide high-resolution estimates of the PDI. Applications for these types of models are found in many different fields of research, e.g., remote sensing (e.g.[53]), meteorology (e.g.[54]), and global health (e.g.[55–60]). These Bayesian frameworks allow, for example, for the incorporation of non-linear covariate effects using Bayesian P-splines[61], or to incorporate complex interactions of covariates, such as two-dimensional surfaces based on the coordinates of the location. Relying on this framework allows us to provide high-resolution maps for the PDI. In particular, we graphically analyze capabilities of physical distancing based on the PDI at the pixel level ($10 \times 10$ km for an analysis of the entire sample, and $5 \times 5$ km for a country-specific analysis). See Section 3.2 of the Supplementary Information for a more thorough discussion of the methodology. The high-resolution maps are shown in this paper for South Africa, Ethiopia, Kenya, and Ghana—the four countries on the African continent that have been the most affected by the COVID-19 pandemic.

The final index is normalized between zero and one, using data on the minimum and maximum values across countries (for national comparisons) or first-level administrative units within countries (for subnational comparisons) and across households. A PDI of one means lowest access to private infrastructure. A PDI of zero indicates a high access to essential home infrastructure.

**Reporting summary**. Further information on research design is available in the Nature Research Reporting Summary linked to this article.

## Data availability

All data sets used in this article are publicly available from the cited sources. The raw data consisting of the DHS data sets and the Socioeconomic Data and Applications Center (SEDAC) gridded population of the world (GPW), v4 data set, are available from the following sources: DHS[62] https://dhsprogram.com/data/; GPW[52] https://sedac.ciesin.columbia.edu/data/collection/gpw-v4. The underlying data which are based on the data sets from the two previous sources and that support the results and conclusion drawn from this study are available upon reasonable request from the corresponding author.

## Code availability

The custom-made **R**-code used to obtain the results has been deposited by the corresponding author and is available upon request.

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

## Acknowledgements

J.S. acknowledges financial support through the research project P-33941 of the Austrian Science Fund (FWF). The computational results presented here have been achieved (in part) using the LEO HPC infrastructure of the University of Innsbruck.

## Author contributions

K.H. and J.S. processed and analyzed the data, interpreted the results. I.G., K.H., and J.S. wrote the first draft of the manuscript. I.G. and J.U. conceptualized the project and revised the manuscript. I.G., K.H., J.S., and J.U. provided important intellectual content. All authors, I.G., K.H., J.S., and J.U., commented on and approved the final version of the paper for submission.

## Competing interests

The authors declare no competing interests.
