## [Peer Review File · Nature Communications]

Reviewer comments, first round -

Reviewer #1 (Remarks to the Author):

This is an interesting paper looking at the availability of household-level infrastructure to protect against infectious diseases such as COVID-19. The authors propose an index (PDI) with 5 indicators, calculate those using DHS data for available countries in Africa at the grid level. The authors show substantial heterogeneity within countries and in particular severe constraints at the household-level that would make physical distancing and lockdowns challenging.

My comments are as follows:

1. The paper's motivation could use some updating, as many of the reasons given in the Introduction seem to be less important now given that vaccinations are now available, and emphasis has moved from using lockdowns as a policy response to increasing access to and usage of COVID-19 vaccines. Is this PDI still useful when lockdowns and extreme social distancing measures may not be part of the policy response portfolio?
2. A fundamental concern is the likeness of this paper to Brown et al., which appears to be very similar in vein to this one. The authors provide on p.5 several reasons why this paper differs from Brown et al.; however, it does not seem as though these are meaningful deviations which would make this paper useful in its own right. For example, how do the results in this paper differ when just a simple average is used; in other words, why is the use of PCA an important extension? A similar comment can be made for weighting by population density – how much difference does this make to the results, outside of placing a higher emphasis on households in urban areas? I would strongly encourage the authors to better distinguish their work.
3. Another source with a similar idea that the authors may find useful to include is: Andrew, Alison, Alex Arman, Britta Augsburg and Ivan Kim Taveras, 2020, "Challenges of Adopting Coronavirus Precautions in Low-Income Countries," Blog Post, Institute of Fiscal Studies, London.
4. A more interesting contribution is the emphasis on sub-country heterogeneity, where policy targeting could have more impact, and the authors on p.7 use several countries such as Kenya and Cote D'Ivoire as examples of where "hot spots" emerge. Figure 2 is particularly interesting. I think a stronger argument could be made if evidence could be provided that these are indeed areas where disproportionately high numbers of cases have emerged. For the most part, most countries in Africa have recorded relatively low numbers of cases, which I don't believe can be explained away by poor testing capacity and it would be useful to see if this can be addressed here (as an example, in South Africa which the authors note have had the highest cases, is the virus more prevalent in areas identified as "hot spots"?).
5. As a follow-up, in the limitations the authors write that PDI does not provide predictions, but can help identify regions where infectious disease may spread. What evidence is there that a) simply targeting poor regions is insufficient, and PDI provides a better indicator for policy makers (presumably PDI and household wealth are quite correlated) and b) why PDI indeed is correlated with infectious disease spread, when the spread of such diseases is highly dependent on epidemiological factors?
6. It appears as though there is some repetition in the conclusion, e.g. the limitations.

Reviewer #2 (Remarks to the Author):

This is a very interesting paper that contributes to a nuanced understanding of the geospatial heterogeneities in COVID-19-related physical distancing across countries in sub-Saharan Africa. Household level infrastructure that enable or constrain physical distancing has evolved over the past decade. The manuscript could have been richer if household level infrastructure variables in the DHS during the past decade were used/mapped. This is imperative to shed light on the temporal heterogeneities. Elucidating spatio-temporal heterogeneities is more robust than focusing only on spatial heterogeneities. However, this is not a fatal flaw.

What other critical variables on physical infrastructure are useful for creating the index but are missing from the DHS?

I have reviewed several hundreds of manuscripts for over 60 international journals. This is the first time I am recommending an article for publication, as is.

Reviewer #3 (Remarks to the Author):

In this paper the authors construct a new index designed to measure the extent to which people are able to physically distance themselves from others, a behavior which is important for preventing the spread of COVID-19 and other airborne infectious diseases. The index is comprised of five items related to hygiene, transportation, communication technology and density. This measure is calculated at the subnational level and results in a mapping of vulnerability at the 10x10km level. The authors observe considerable within-country variation in this physical distancing index, and that the greatest constraints to physical distancing vary by region and country (e.g. water in Rwanda and Burundi).

How useful is this index? The authors explicitly note that the index is not meant to be used for predictive purposes at the national level but can be used for targeting resources at the subnational level, such as vaccines in the case of COVID-19. The predictive power even at the local level is hampered by the fact that, as the authors note, that individual and community behavior is important for preventing the spread of disease. Being able to physically distance does not necessarily mean that individuals will practice physical distancing.

Although the authors include many caveats in how the index may be used, and describe this as a descriptive exercise, it would be helpful to a) get a sense of whether there is any relationship at all between this index and actual case loads at the subnational level (they explicit say it is not meant to be predictive at the national level but imply it could be at the subnational level), b) how rapidly measures included in the index change over time, and c) provide a framework/method for adapting the index to other disease if and then they arise. I discuss each of these in turn.

First, although subnational data on caseloads are going to be impossible to acquire for many countries, it seems there must be a few (South Africa being one) where these could be acquired and employed for analysis of the relationship between the index and COVID-19 spread and caseloads. Despite the authors' caveat that the index is not meant to be predictive (at the national level), if there is no relationship between actual caseloads and the index at the subnational level it becomes more difficult to make the case that it would be useful from a policy standpoint. It is also not clear if the authors are suggesting other policy implications beyond vaccination prioritization. If the authors

find a relationship between the index and caseloads at the subnational level, this should be reported. If not, they should explain the conditions under which the index is useful for making policy decisions, and for which types of policy decisions.

Second, the authors explain that they have used the most recent DHS survey for each country in the sample, and that in practice this covers a time period from 2005 to 2018. I am interested in knowing how time variant this index is for two reasons. First, if this index were to be employed in the future, a highly time variant measure would suggest that it should be recalculated with the most recent surveys available at the time, and that the findings from this particular set of surveys may indicate a quite different geographic distribution of vulnerability than in the past or in the future. Second, if the index is variable over time, I am not sure how to think about the inclusion of surveys from 15 or even 10 years ago. Some discussion is merited to justify the inclusion of these surveys if indeed the index varies over time. While the DHS is not conducted consistently everywhere, there are some countries where it has been regularly conducted and this subsample could be employed to assess the stability of the index within a given country over time.

Third, this particular index was (understandably) developed for the specific case of COVID-19, which has, like any infectious disease, a set of specific characteristics that affect its spread. As such, physical distancing was selected as a behavior to focus on, and measures relating to the ability to practice physical distancing included in the present index. At the same time, though, authors frequently refer to infectious diseases in general, suggesting that the index will/is useful not only for the present pandemic but also future outbreaks. We know with certainty there will be such outbreaks, although the features of the infectious disease are likely to differ. Past experience suggests that Ebola is likely to emerge in Africa again in the near future, for example. I would like to see a more extended discussion on how/whether the index provided here can either be deployed or revised to apply to future outbreaks, and a framework for thinking about how to do this. To take the Ebola case, are there other measures that would make more sense to include in the index of infrastructural vulnerability for the spread of Ebola? Are there some in the present index that should be removed? What are the factors that should be considered in deciding which measures go into the index? In other words, the contribution of the paper would be stronger if the authors provide not a one-time index – of a specific time period for a specific disease (which may or may not correlate with actual caseloads even of that disease, as discussed above) but also a framework for thinking about how to use the methodology to generate indices for future outbreaks of infectious diseases, and the dimensions that would go into selecting the most appropriate measures (for example, a) how infection is spread – airborne, bodily fluids, etc., b) how long the virus/bacteria etc. persists in air or on surfaces, c) the basic reproduction number of the infectious disease, etc).

Other notes/questions:

How sensitive is the index to the inclusion/removal of each component? If it is very sensitive, i.e. the pattern of vulnerability looks quite different if one element is removed, this raises important questions about the selection of measures and demonstrates how carefully this would need to be done in the case of future infectious diseases.

There are a number of typos. For example, p. 5 “household hast internet...”, “principle component analysis is sed for the PDI”.

The sentence “We graphically analyse capabilities of physical distancing based on the PDI at the

national level (34 countries), at a sub-national regional level (519 regions) as well as high-resolution, pixel level (5 x km)." is repeated in the first full paragraph on page 5.

Should we think differently about shared spaces that are indoor (households or transport) than outdoor (water points)? It seems like in the case of COVID the indoor shared spaces are a greater risk.

Response to Referees for the Manuscript Entitled, *Manuscript Entitled, When physical distancing becomes impossible: an index based on access to essential infrastructure (NCOMMS-21-38809)* submitted to **Nature Communications**

February 17, 2022

Reviewer Comments.

Reviewer #1

This is an interesting paper looking at the availability of household-level infrastructure to protect against infectious diseases such as COVID-19. The authors propose an index (PDI) with 5 indicators, calculate those using DHS data for available countries in Africa at the grid level. The authors show substantial heterogeneity within countries and in particular severe constraints at the household-level that would make physical distancing and lockdowns challenging.

Comment 1 The paper’s motivation could use some updating, as many of the reasons given in the Introduction seem to be less important now given that vaccinations are now available, and emphasis has moved from using lockdowns as a policy response to increasing access to and usage of COVID-19 vaccines. Is this PDI still useful when lockdowns and extreme social distancing measures may not be part of the policy response portfolio

Response We thank Reviewer #1 for this valuable comment. We agree that since the submission of our piece, the dynamics of the COVID-19 pandemic have changed considerably, and hence, we updated the motivation of our study accordingly. However, we argue that the relevance of a physical distancing index (PDI) did not lose its momentum, neither for the COVID-19 pandemic, nor for other infectious diseases that are highly transmissible. First, even if SARS-CoV-2 case numbers have been lower for the African continent than initially expected, there is an underestimation in low-income countries, both in terms of case numbers (explained by the lack of testing facilities), and fatalities, as revealed by analyzing a country’s excess mortality (reported mortality - expected mortality) [1].

Second, SARS-CoV-2 has undergone various mutations. The current wave is largely dominated by the emergence and rapid spread of the Omicron variant, which shows enhanced transmissibility and considerable immune evasion [2]. As a result, most countries still have policies of physical distancing in place, even if their populations may not be able to follow them.

Third, it is almost certain that SARS-CoV-2 will continue to mutate, and therefore, physical distancing will remain an important non-pharmaceutical intervention. In particular, the currently low vaccination coverage in Africa has been discussed as a driver of future mutations of the virus [3]. For example, while about 48.5% of the global population is fully vaccinated against COVID-19, only about 4.3% of the population in low-income countries is fully vaccinated as of the end of December 2021. Particularly low vaccination rates are observed in Nigeria with only 1.3% or Ethiopia with 2.5% [4, 5]. It follows that although multiple vaccines exist, access is still limited for most low-income countries and measures of physical distancing remain important (or necessary) policy responses.

Finally, many epidemiologists and public health specialists argue that we will have to deal with an increasing number of emerging and re-emerging diseases, many of which are caused by viruses that might jump from animals to humans [6, 7, 8]. Hence, the development of the spread of COVID-19 as well as decreasing the risk of future

variants will, in the short term, strongly depend on the progress of national and global vaccination campaigns. In the long term, it will depend on investments in infrastructure that allow households to reduce risks of getting infected in the first place—otherwise poorer countries will have to again resort to lengthy school closures that are easiest in the short term, but have devastating effects in the long term [9, 10].

Taken together, we feel that our PDI is useful because it shows, at a subnational level, where infrastructural constraints are a particular concern, resulting in a higher risk of an outbreak or hotspots of new infections. This knowledge could be relevant for policy makers trying to distribute vaccines more effectively. In addition, it is useful for target monitoring in the short term and could motivate policy makers to prioritize infrastructure investment in the long term.

In addition to these arguments, we now also highlight the applicability of our approach as a general framework to identify and calculate a similar index for future epidemics on page 12.

Comment 2 A fundamental concern is the likeness of this paper to Brown et al., which appears to be very similar in vein to this one. The authors provide on p.5 several reasons why this paper differs from Brown et al.; however, it does not seem as though these are meaningful deviations which would make this paper useful in its own right. For example, how do the results in this paper differ when just a simple average is used; in other words, why is the use of PCA an important extension? A similar comment can be made for weighting by population density—how much difference does this make to the results, outside of placing a higher emphasis on households in urban areas? I would strongly encourage the authors to better distinguish their work.

Response Our study has four fundamental differences compared to the recent piece by Brown and colleagues [11], which we now highlight in the paper in more detail in section four and in the results section:

First, Brown and colleagues [11] use different indicators. Similar to us, they also use information on access to the internet or a mobile phone, private water as well as sanitation, and the number of people per room. In contrast to us, they use information on whether a household has a place for handwashing with soap, which we argue is more a measure of hygiene than a measure of physical distancing capacity. Another indicator we use, but they don't, is whether the household has a means of private transportation, which we argue is also an important measure of the feasibility of physical distancing measures.

Second, we exploit the availability of geo-referenced information in the Demographic and Health Surveys (DHS) to provide new insights about the capability of physical distancing at the subnational level. Brown et al. [11] provide their results only at the country level. Although this is important, we argue that it is of particular interest and importance to provide information about the within-country heterogeneity of potential high-risk areas because this allows a much more effective and detailed policy response to contain the spread of infectious diseases, a strategy that many high-income countries also followed in their measures to identify hotspots of new COVID-19 infections through contact tracing.

Third, Brown and colleagues [11] use a simple country-average of their indicators to calculate their index. Although this is a straightforward approach, it also implies an arbitrary weighting scheme, where one has to assume that, for example, access to the internet has the same informative power as having access to piped drinking water when explaining the capacities of households to protect themselves from getting infected. Therefore, we employ the principal component analysis (PCA) method to avoid this equal weighting assumption. The PCA is a more data-driven approach and it combines the variation of all included variables in the index. Compared to a (equally) weighted average, this procedure 'weights' the variables according to their contribution to the 'overall variation'. Accordingly, it is less arbitrary. This is also nicely illustrated in Fig. SI 2.2 (a) that shows the explained variance by components of the index. The figure clearly shows that each indicator does not have an equal weight in calculating the index. Hence, we argue that a PCA is a more suitable method to calculate an index that provides information on household capability to engage in social distancing than taking a simple average across indicators.

Fourth, another fundamental difference is to take into account population density, which we argue is crucial in studying the capabilities of households for physical distancing. Clearly, population density should be a main determinant of infection rates within countries if the infrastructure needed to keep distance is missing. Hence, not taking into account population density masks a lot of information within both urban and rural areas that can be useful for a better identification of disease hotspots and also for a better targeted distribution of vaccines.

Therefore, adding geo-referenced information on population density allows us to produce nuanced and meaningful insights into the capabilities of physical distancing within countries. This is also illustrated in the new Fig. 3 that we added to our manuscript, which shows the differences between the weighted PDI and the actual caseload (see also the response to comment 4). The figure below shows the differences between an unweighted PDI and the population-weighted PDI in comparison to the actual caseload in South Africa. Two major aspects are noticeable: First, the figure highlights, again, the predictive power of our index in general (see also the detailed response to the Reviewer’s fourth comment on the predictive power of the index within countries). Second, the map shows that taking into account the population density is particularly important to improve the predictive power of the index. The correlation coefficient between the unweighted PDI and COVID-19 cases is 0.47, whereas it is 0.84 for the weighted PDI.

PDI, population weighted PDI and observed caseload at the regional level for South Africa. Note that we used the latest available information on the aggregated regional caseload. *Source:* Information on the regional caseload for South Africa is publicly available data which we are happy to share upon request; calculations by the authors.

The aforementioned differences in the method to calculate an index of physical distancing result in differences in the findings between our study and the study by Brown and colleagues. The correlation coefficient between the index by Brown et al. [11] and our index at the country level is only -0.21 (note that it is negative because weaker protection has a higher value in our index, whereas it has a lower value in Brown’s index). The low association between the two indices also results in a very different ranking of countries across indices. Moreover, our index shows much more variance than the one from Brown and colleagues [11]. These differences are illustrated in Fig. 5 in the main text (see respective figure below), which shows the scatterplot of the PDI versus the home environment for protection index (HEP). These differences are now also discussed and highlighted in more detail in the results section of our study.

Scatter plot of PDI (using the latest available Demographic Health Survey of a country) against the home environment for protection index. Note that a high PDI implies a lack of essential private infrastructure, and hence, high risk of disease transmission; accordingly, countries in the top of boxes have a lack of private infrastructure, and hence, lack the capacity to limit disease transmission. Moreover, note that for the infectious disease vulnerability index, the WHO international health regulations index, and the home environment for protection index, a higher value indicates a better preparedness, or protection. *Source:* DHS, Moore and colleagues [12], Gilbert et al. [13], and World Development Indicators [14]; calculations by the authors.

Comment 3 Another source with a similar idea that the authors may find useful to include is: Andrew, Alison, Alex Arman, Britta Augsburg and Ivan Kim Taveras, 2020, “Challenges of Adopting Coronavirus Precautions in Low-Income Countries,” Blog Post, Institute of Fiscal Studies, London.

Response Thank you for pointing us to this interesting blog post, which is indeed broadly related to our study and and to Brown et al. [11]. Yet, similar to Brown and colleagues [11], this blog post only looks at the capacity of households to protect household members from SARS-CoV-2 at the country level, while it does not take into account population density. We now provide a cross-reference to this blog post in the section where we introduce prior studies with similar approaches that have been discussed in the literature (see page 3).

Comment 4 A more interesting contribution is the emphasis on sub-country heterogeneity, where policy targeting could have more impact, and the authors on p.7 use several countries such as Kenya and Cote D’Ivoire as examples of where “hot spots” emerge. Figure 2 is particularly interesting. I think a stronger argument could be made if evidence could be provided that these are indeed areas where disproportionately high numbers of cases have emerged. For the most part, most countries in Africa have recorded relatively low numbers of cases, which I don’t believe can be explained away by poor testing capacity and it would be useful to see if this can be addressed here (as an example, in South Africa which the authors note have had the highest cases, is the virus more prevalent in areas identified as “hot spots”?).

Response Thank you for this excellent comment. We agree that we can make a stronger argument for the usage of the PDI if we can provide evidence that the value of the PDI is correlated with actual case numbers

within countries. To analyze the usefulness of our approach and our index, we checked if data on reported cases are available at the subnational level for each country in our sample. We identified nine countries with subnational information: Democratic Republic of the Congo, Ethiopia, Mozambique, Namibia, Nigeria, Niger, Senegal, South Africa, and Togo. Unfortunately these numbers are only available at a regional level and not at a 10 x 10 km level. Hence, we can only check for correlations between COVID-19 case numbers and the PDI at the regional level. To analyze the correlation between the PDI and the actual caseload, we used the most recent data available on confirmed cumulative COVID-19 cases. For these nine countries, we calculated the correlation coefficient between the regional caseload and the regional PDI (weighted by the population density, see our response to comment 2). The correlation coefficient ranges from 0.4 to 0.9, pointing to a close association between our index and the actual caseload. This high correlation shows the usefulness of the PDI to identify hotspots within a country, and how limited infrastructure and densely populated areas are closely related to the observed caseload. To illustrate the predictive power of the PDI, we now include and describe Fig. 3 in the manuscript that shows the regional PDI compared to the actual caseload at the regional level for South Africa. The comparisons for the other eight countries are shown in the supplement in Fig. SI 1.3 and Fig. SI 1.4.

Population weighted PDI and observed caseload at the regional level for South Africa. Note that we used the latest available information on the aggregated regional caseload. *Source:* Information on the regional caseload for South Africa is publicly available data which we are happy to share upon request and DHS; calculations by the authors.

However, we would also like to emphasize that obtaining numbers on the actual caseload within countries is difficult and the numbers need to be treated with caution. It requires that the caseload is either based on representative samples of the population, or that the caseload is directly calculated from the population (which is extremely unlikely in low-income countries). Using confirmed cases will introduce some bias because of underreporting, which is likely to be higher in rural areas and hence the correlation could even be higher. Therefore, we did not want to put too much emphasis on the relationship between the index and the observed caseload. We still point out that the close association of our index and the actual number of confirmed cases provides a strong argument about the usefulness of the PDI to identify hotspots of potential outbreaks.

Comment 5 As a follow-up, in the limitations the authors write that PDI does not provide predictions, but can help identify regions where infectious disease may spread. What evidence is there that a) simply targeting poor regions is insufficient, and PDI provides a better indicator for policy makers (presumably PDI and household wealth are quite correlated) and b) why PDI indeed is correlated with infectious disease spread, when the spread of such diseases is highly dependent on epidemiological factors?

Response a) Thank you for this comment. It is indeed plausible to argue that poverty and the index are closely related and that poverty is hence a good indicator of the spread of SARS-CoV-2 and that the PDI does not add any additional information. We now discuss this issue on page 11 in the manuscript and include the graph below as Fig. SI 1.5 in the manuscript.

First, Fig. 5 in the main manuscript shows that there is a relationship between the PDI and GDP per capita at the country level.

Second, on a regional level (see figure below), we already see that poverty alone is not a very good predictor of potential disease hotspots. For example, rural areas are, on average, poorer than urban areas in Africa. However, we see that our index is not necessarily higher in rural areas because we take population density into account (this is also different to the study put forth by Brown et al. [11], see our response to comment 2).

Third, unfortunately, the availability of data on poverty within countries is very limited. However, to analyze the correlation between poverty and the PDI, we used regional data on the share of people living below the national poverty line from Mozambique, Namibia, Nigeria, and South Africa. There is, in all cases, only a low correlation between the poverty headcount rate and the PDI. Also a linear model suggests only a minor relationship using the adjusted R^2 .

Taken together, we argue that poverty alone is not a good predictor of potential disease hotspots and that a more specific index is needed to identify potential high-risk areas

Scatterplot of the population weighted PDI at the regional level and the poverty headcount rate at the regional level for countries where subnational estimates of the poverty headcount rate is available. These countries are Mozambique, Namibia, Nigeria, and South Africa. *Source:* DHS and National Statistical Offices; calculations by the authors.

b) As described in the response to the previous comment, at this stage of the COVID-19 pandemic, it is possible to use an ex-post perspective to investigate whether there is a relationship between the PDI and the observed caseload at the subnational level. We find a high relationship between the PDI and the observed subnational caseload. Importantly, this observation is only based on the countries for which subnational caseload data exist. This can be interpreted to mean that crucial infrastructure is necessary to allow physical distancing, and hence, to contain the spread of COVID-19, despite epidemiological factors. Across a specific country

both predictions should not be done. As already indicated in the previous version of our manuscript, country differences in policies, general behavior of the population, and exposure to imported COVID-19 cases are also highly relevant for the spread of the disease and vary widely across countries (but less so within countries). Hence, across countries, the PDI can be used as a benchmark for how well countries dealt with the pandemic.

Comment 6 It appears as though there is some repetition in the conclusion, e.g. the limitations.

Response Thank you for this comment. We carefully checked and revised the conclusion in order to delete any existing repetitions and redundancies.

Reviewer #2

This is a very interesting paper that contributes to a nuanced understanding of the geospatial heterogeneities in COVID-19-related physical distancing across countries in sub-Saharan Africa. Household level infrastructure that enable or constrain physical distancing has evolved over the past decade.

The manuscript could have been richer if household level infrastructure variables in the DHS during the past decade were used/mapped. This is imperative to shed light on the temporal heterogeneities. Elucidating spatio-temporal heterogeneities is more robust than focusing only on spatial heterogeneities. However, this is not a fatal flaw.

I have reviewed several hundreds of manuscripts for over 60 international journals. This is the first time I am recommending an article for publication, as is.

Response We are grateful to Reviewer #2 for the overall positive appraisal of our research and the very encouraging words.

Comment 1 What other critical variables on physical infrastructure are useful for creating the index but are missing from the DHS?

Response That is indeed an interesting question. There are several variables that could play an important role here, such as the type of work people do in order to analyze whether working from home is even an option or not. For example, one could look at the share of people working in agriculture, trade, or office work. However, information on these indicators would reduce the sample size significantly given the data at hand. We therefore decided not to take this information into account.

Reviewer #3

In this paper the authors construct a new index designed to measure the extent to which people are able to physically distance themselves from others, a behavior which is important for preventing the spread of COVID-19 and other airborne infectious diseases. The index is comprised of five items related to hygiene, transportation, communication technology and density. This measure is calculated at the subnational level and results in a mapping of vulnerability at the 10x10km level. The authors observe considerable within-country variation in this physical distancing index, and that the greatest constraints to physical distancing vary by region and country (e.g. water in Rwanda and Burundi).

How useful is this index? The authors explicitly note that the index is not meant to be used for predictive purposes at the national level but can be used for targeting resources at the subnational level, such as vaccines in the case of COVID-19. The predictive power even at the local level is hampered by the fact that, as the authors note, that individual and community behavior is important for preventing the spread of disease. Being able to physically distance does not necessarily mean that individuals will practice physical distancing.

Although the authors include many caveats in how the index may be used, and describe this as a descriptive exercise, it would be helpful to a) get a sense of whether there is any relationship at all between this index and actual case loads at the subnational level (they explicit say it is not meant to be predictive at the national level but imply it could be at the subnational level), b) how rapidly measures included in the index change over time, and c) provide a framework/method for adapting the index to other disease if and then they arise. I discuss each of these in turn.

Comment 1 First, although subnational data on caseloads are going to be impossible to acquire for many countries, it seems there must be a few (South Africa being one) where these could be acquired and employed for analysis of the relationship between the index and COVID-19 spread and caseloads. Despite the authors' caveat that the index is not meant to be predictive (at the national level), if there is no relationship between actual caseloads and the index at the subnational level it becomes more difficult to make the case that it would be useful from a policy standpoint. It is also not clear if the authors are suggesting other policy implications beyond vaccination prioritization. If the authors find a relationship between the index and caseloads at the subnational level, this should be reported. If not, they should explain the conditions under which the index is useful for making policy decisions, and for which types of policy decisions.

Response We thank Reviewer #3 for this excellent comment, which has also been raised by another Reviewer. We agree with the comment that we can make a stronger argument for the usage of the PDI if we can provide evidence that the value of the PDI is associated with actual cases of COVID-19. We also agree that it can only be predictive at the subnational level and not the national level, where differences in policies and people's behavior or populations across countries are too large.

To analyze the usefulness of our approach and our index, we checked, for every country in our sample, if data on reported cases is available at the subnational level, and identified nine countries with subnational information. The countries are: Democratic Republic of the Congo, Ethiopia, Mozambique, Namibia, Nigeria, Niger, Senegal, South Africa, and Togo. Unfortunately these numbers are only available at a regional level and not at a 10 x 10 km level. Hence, we can only check for correlations between COVID-19 case numbers and a PDI at the regional level. For analyzing the correlation of the PDI with the actual caseload, we used the most recent data available on confirmed cumulative COVID-19 cases. For these nine countries, we calculated the correlation coefficient between the regional caseload and the regional PDI (weighted by the population density). The correlation coefficient ranges from 0.4 to 0.9, pointing to an overall close association between our index and the actual caseload. This high correlation shows the usefulness of the PDI to identify hotspots within a country, and how limited infrastructure and densely populated areas are closely related to the observed caseload. To illustrate the predictive power of the PDI, we now include and describe Fig. 3 in the manuscript, which shows the regional PDI compared to the actual caseload at the regional level for South Africa (see figure below). The comparisons for the other eight countries are shown in the supplement in Fig. SI 1.3 and Fig. SI 1.4.

Population weighted PDI and observed caseload at the regional level for South Africa. Note that we used the latest available information on the aggregated regional caseload. *Source:* Information on the regional caseload for South Africa is publicly available data which we are happy to share upon request; calculations by the authors.

However, we would also like to emphasize that obtaining numbers on the actual caseload within countries is difficult and the numbers need to be treated with caution. It requires that the caseload is either based on representative samples of the population, or that the caseload is directly calculated from the population (which is extremely unlikely in low-income countries). Using confirmed cases will introduce some bias because of underreporting, which is likely to be higher in rural areas, and hence, the correlation could even be higher. Therefore, we did not want to put too much emphasis on the relationship between the index and the observed caseload. We still point out that the close association of our index and the actual number of confirmed cases provides a strong argument about the usefulness of the PDI to identify hotspots of potential outbreaks.

Comment 2 Second, the authors explain that they have used the most recent DHS survey for each country in the sample, and that in practice this covers a time period from 2005 to 2018. I am interested in knowing how time variant this index is for two reasons. First, if this index were to be employed in the future, a highly time variant measure would suggest that it should be recalculated with the most recent surveys available at the time, and that the findings from this particular set of surveys may indicate a quite different geographic distribution of vulnerability than in the past or in the future. Second, if the index is variable over time, I am not sure how to think about the inclusion of surveys from 15 or even 10 years ago. Some discussion is merited to justify the inclusion of these surveys if indeed the index varies over time. While the DHS is not conducted consistently everywhere, there are some countries where it has been regularly conducted and this subsample could be employed to assess the stability of the index within a given country over time.

Response Thank you very much for this useful suggestion. We agree that a trend analysis of the index is very interesting to test the stability of the index over time. In the revised version of the manuscript, we now present the results for the PDI for all country- and survey-year combinations as Fig. SI 1.2 (see figure below) and point towards a more in depth interpretation of the dynamics of the index over time in the main text on page 4. The graph points to an overall stability of the index with only slow changes over time and only minor changes in the ranking across countries. However, some countries show a slight increase in the index (e.g., Mozambique), while others experience a slight decrease over time (e.g., Ghana), with decreasing values pointing towards an overall improvement in infrastructure. However, our calculations for the most recent available survey year do not go

back further in time than 2-6 years for most countries, except for Swaziland (2007), Cameroon (2011), and Côte d’Ivoire and Gabon (2012). Hence, we argue that our index provides a timely estimate of a country’s capacity for physical distancing. However, we are also happy to leave out these four countries, if this is requested by Reviewer #3.

Development of the PDI over time for all countries with complete information. Note that the PDI is normalized between zero and one for the whole time period; calculations by the authors.

Comment 3 Third, this particular index was (understandably) developed for the specific case of COVID-19, which has, like any infectious disease, a set of specific characteristics that affect its spread. As such, physical distancing was selected as a behavior to focus on, and measures relating to the ability to practice physical distancing included in the present index. At the same time, though, authors frequently refer to infectious diseases in general, suggesting that the index will/is useful not only for the present pandemic but also future outbreaks. We know with certainty there will be such outbreaks, although the features of the infectious disease are likely to differ. Past experience suggests that Ebola is likely to emerge in Africa again in the near future, for example.

I would like to see a more extended discussion on how/whether the index provided here can either be deployed or revised to apply to future outbreaks, and a framework for thinking about how to do this. To take the Ebola case, are there other measures that would make more sense to include in the index of infrastructural vulnerability for the spread of Ebola? Are there some in the present index that should be removed? What are the factors that should be considered in deciding which measures go into the index? In other words, the contribution of the paper would be stronger if the authors provide not a one-time index – of a specific time period for a specific disease (which may or may not correlate with actual caseloads even of that disease, as discussed above) but also a framework for thinking about how to use the methodology to generate indices for future outbreaks of infectious diseases, and the dimensions that would go into selecting the most appropriate measures (for example, a) how infection is spread – airborne, bodily fluids, etc., b) how long the virus/bacteria etc. persists in air or

on surfaces, c) the basic reproduction number of the infectious disease, etc).

Response Thank you for this excellent comment. Although the proposed PDI was motivated by the ongoing COVID-19 pandemic, we agree that our approach can provide a basis for a more general framework to be applied for future outbreaks of other infectious diseases, though context, disease agent, transmissibility and human behavior will play important roles. We highlight this now in the conclusion on page 12.

To make use of a more generic index and the available data to identify potential hotspots of outbreaks of future diseases, it is necessary to: first, identify epidemiological factors that help to contain the spread of a disease. Second, identify relevant non-pharmaceutical measures that help to contain the spread of this disease. Third, find and identify relevant variables that correspond to these measures and are included in the standard data sources, e.g., for sub-Saharan Africa and many low- and middle-income countries these are the Demographic and Health Surveys (DHS) and the Multiple Indicator Cluster Surveys (MICS). Fourth, calculate a one-dimensional index that overlaps with the characteristics of the spread of the disease. Fifth, take advantage of the geo-referenced nature of such surveys and map the created index. Sixth, validate the usefulness of the index, ex-post, and the drawn conclusions.

For the still ongoing COVID-19 pandemic, at the current stage of the pandemic, it is possible to use a kind of ex-post perspective to investigate whether there is a relationship between the PDI and the observed caseload. We find a surprisingly high relationship between the PDI and the observed subnational caseload (of course this observation is only based on the countries for which subnational caseload data exist). This can be interpreted to mean that certain infrastructure contained in the PDI is necessary to contain the spread of COVID-19.

Other notes/questions

Comment 4 How sensitive is the index to the inclusion/removal of each component? If it is very sensitive, i.e. the pattern of vulnerability looks quite different if one element is removed, this raises important questions about the selection of measures and demonstrates how carefully this would need to be done in the case of future infectious diseases.

Response Thank you for raising this important aspect. As a sensitivity analysis, we now calculate several distinct variants of the index, omitting different components. We found that the correlation of the different variants was always very high. This highlights the robustness and small sensitivity of the index to the omission of individual components when calculating the index. The exception being population density—for example, as expected, urban areas are usually more densely populated, but also show better infrastructure. However, as the first figure below shows, including population density increases the predictive power of the PDI at the subnational level. In addition, this emphasizes the advantage of a method for dimension reduction, such as PCA, over an index that is simply the weighted average of the individual components. Furthermore, the included variables are selected based on the WHO recommendations to contain the spread of SARS-CoV-2. We now include Fig. SI 2.1 (see figure below) in the description of the methodology in the supplementary materials and mention this issue on page 5.

PDI, population weighted PDI and observed caseload at the regional level for South Africa. Note that we used the latest available information on the aggregated regional caseload. *Source:* Information on the regional caseload for South Africa is publicly available data which we are happy to share upon request; calculations by the authors.

Sensitivity analysis of the index at the country level. Correlation of the unweighted PDI and several variants of the PDI with the population weighted PDI. *Source:* DHS; calculations by the authors.

Comment 5 There are a number of typos. For example, p. 5 “household hast internet...”, “principle component analysis is set for the PDI”.

Response We carefully revised the entire manuscript to remove typos and asked an English native speaker to proofread the manuscript. Of course, any errors remain our own.

Comment 6 The sentence “We graphically analyse capabilities of physical distancing based on the PDI at the national level (34 countries), at a sub-national regional level (519 regions) as well as high-resolution, pixel level (5 x 5 km).” is repeated in the first full paragraph on page 5.

Response Thanks for highlighting this redundancy. We corrected this and carefully checked the rest of the manuscript for similar occurrences.

Comment 7 Should we think differently about shared spaces that are indoor (households or transport) than outdoor (water points)? It seems like in the case of COVID the indoor shared spaces are a greater risk.

Response That is a very interesting question. The differentiation between indoor and outdoor shared spaces is important to analyze, especially for the COVID-19 pandemic, for which it has been shown that transmission indoors is much higher than transmission outdoors (with the same protective measures). See, for example, [15, 16, 17]. For our indicators, we can differentiate between what is indoor and outdoor: water sharing is most likely outdoor whereas, everything else is indoor; even though two people would not be simultaneously in one shared sanitation space, sanitation could also be considered to be outdoor. We still decided to keep all indicators in our index, also because results do not change much if we take out individual indicators (see our response to comment 4). If the reviewer still thinks this is necessary, we can take out access to private water in the final version of the manuscript. We describe this point on page 5 in the manuscript.

References

- [1] Karlinsky, A. & Kobak, D. Tracking excess mortality across countries during the COVID-19 pandemic with the world mortality dataset. *eLife* **10**, e69336 (2021). DOI: 10.7554/eLife.69336.
- [2] Rzymiski, P. et al. Covid-19 vaccine boosters: The good, the bad, and the ugly. *Vaccines* **9** (2021). DOI: 10.3390/vaccines9111299.
- [3] Iftekhhar, E. N. et al. A look into the future of the COVID-19 pandemic in Europe: an expert consultation. *Lancet Regional Health – Europe* **8**, 100185 (2021). DOI: 10.1016/j.lanpe.2021.100185.
- [4] Our World in Data. Share of the population fully vaccinated against COVID-19. Data set, Our World in Data (2021). <https://github.com/owid/covid-19-data/tree/master/public/data/vaccinations>.
- [5] Mathieu, E. et al. A global database of COVID-19 vaccinations. *Nature Human Behaviour* **5**, 947–953 (2021). DOI: 10.1038/s41562-021-01122-8.
- [6] Grange, Z. L. et al. Ranking the risk of animal-to-human spillover for newly discovered viruses. *Proceedings of the National Academy of Sciences* **118** (2021). DOI: 10.1073/pnas.2002324118.
- [7] Gomez, G. B., Mahé, C. & Chaves, S. S. Uncertain effects of the pandemic on respiratory viruses. *Science* **372**, 1043–1044 (2021). DOI: 10.1126/science.abh3986.
- [8] Donnik, I. M. et al. Coronavirus infections of animals: Future risks to humans. *Biology Bulletin* **48**, 26–37 (2021). DOI: 10.1134/S1062359021010052.
- [9] Gros, C., Valenti, R., Schneider, L., Valenti, K. & Gros, D. Containment efficiency and control strategies for the corona pandemic costs. *Scientific Reports* **11**, 6848 (2021). DOI: 10.1038/s41598-021-86072-x.
- [10] Sanz-Muñoz, I., Tamames-Gómez, S., Castrodeza-Sanz, J., Eiros-Bouza, J. M. & de Lejarazu-Leonardo, R. O. Social distancing, lockdown and the wide use of mask; a magic solution or a double-edged sword for respiratory viruses epidemiology? *Vaccines* **9** (2021). DOI: 10.3390/vaccines9060595.
- [11] Brown, C. S., Ravallion, M. & van de Walle, D. Can the world’s poor protect themselves from the new coronavirus? Working Paper 27200, National Bureau of Economic Research, Cambridge, Massachusetts (2020). Preprint at <https://doi.org/10.3386/w27200>.
- [12] Moore, M., Gelfeld, B., Okunogbe, A. T. & Paul, C. Identifying future disease hot spots: Infectious disease vulnerability index. Tech. Rep., RAND Corporation, Santa Monica, California (2016). DOI: 10.7249/RR1605.
- [13] Gilbert, M. et al. Preparedness and vulnerability of African countries against importations of COVID-19: a modelling study. *Lancet* **395**, 871–877 (2020). DOI: 10.1016/S0140-6736(20)30411-6.
- [14] World Bank. World Development Indicators 2020. Data set, World Bank Group, Washington, D.C. (2020). <https://data.worldbank.org/>.
- [15] Bhagat, R. K., Davies Wykes, M. S., Dalziel, S. B. & Linden, P. F. Effects of ventilation on the indoor spread of COVID-19. *Journal of Fluid Mechanics* **903**, F1 (2020). DOI: 10.1017/jfm.2020.720.
- [16] Rowe, B., Canosa, A., Drouffe, J. & Mitchell, J. Simple quantitative assessment of the outdoor versus indoor airborne transmission of viruses and COVID-19. *Environmental Research* **198**, 111189 (2021). DOI: 10.1016/j.envres.2021.111189.
- [17] Senatore, V. et al. Indoor versus outdoor transmission of SARS-CoV-2: environmental factors in virus spread and underestimated sources of risk. *Euro-Mediterranean Journal for Environmental Integration* **6**, 30 (2021). DOI: 10.1007/s41207-021-00243-w.

Reviewer comments, second round -

Reviewer #1 (Remarks to the Author):

The authors have addressed all the comments I had satisfactorily. I would be fine with the reviewed version of this paper proceeding for publication.

Reviewer #2 (Remarks to the Author):

My original comments have been satisfactorily addressed by the authors. I recommend that the manuscript be accepted.

Reviewer #3 (Remarks to the Author):

I thank the authors for their efforts and attentiveness to the comments raised by the reviewers and am satisfied with their additions and responses.

My one outstanding comment is that I think it does a disservice to important descriptive work to list the "descriptive nature" of the study as a limitation. I think this "limitation" can be removed.

Response to Referees for the Manuscript Entitled, *An index of access to essential infrastructure to identify where physical distancing is impossible* (NCOMMS-21-38809A) submitted to **Nature Communications**

April 8, 2022

Reviewer Comments.

Reviewer #1

The authors have addressed all the comments I had satisfactorily. I would be fine with the reviewed version of this paper proceeding for publication.

Response We thank Reviewer #1 for the overall positive feedback and the encouraging words during the entire reviewing process.

Reviewer #2

My original comments have been satisfactorily addressed by the authors. I recommend that the manuscript be accepted..

Response We are grateful to Reviewer #2 for the overall positive appraisal of our research and the very encouraging words.

Reviewer #3

I thank the authors for their efforts and attentiveness to the comments raised by the reviewers and am satisfied with their additions and responses.

Response We thank Reviewer #3 for refereeing the manuscript and helping to substantially improve the content of the article throughout the whole reviewing process.

Comment 1 My one outstanding comment is that I think it does a disservice to important descriptive work to list the "descriptive nature" of the study as a limitation. I think this "limitation" can be removed.

Response Thank you very much for your positive feedback. As suggested the part we mention the descriptive nature of the study as limiting factor has been removed from the limitations.